Comparative assessment of single-stage and two-stage anaerobic digestion for biogas production from high moisture municipal solid waste

Markphan Wattananarong 1
Mamimin Chonticha 2
Suksong Wantanasak 3
Prasertsan Poonsuk 2
O-Thong Sompong sompong.o@gmail.com 4 5
1 Environmental Program, Faculty of Sciences and Technology, Nakhon Si Thammarat Rajabhat University , Nakhon Si Thammarat , Thailand
2 Research and Development Office, Prince of Songkla University , Songkhla , Thailand
3 School of Bioresources and Technology, King Mongkut’s University of Technology Thonburi , Bangkok , Thailand
4 International College, Thaksin University , Songkhla , Thailand
5 Sustainable Agricultural Resources Management Program, Faculty of Technology and Community Development, Thaksin University , Phatthalung , Thailand
Marques Marcia
Electronic publication date: 2020 Aug 19
Publication date: 2020
Volume: 8
Electronic Location ID: e9693
Received 2019 Oct 13; Accepted 2020 Jul 20
Copyright: ©2020 Markphan et al.
Copyright year: 2020
Copyright holder: Markphan et al.
License: This is an open access article distributed under the terms of the Creative Commons Attribution License, which permits unrestricted use, distribution, reproduction and adaptation in any medium and for any purpose provided that it is properly attributed. For attribution, the original author(s), title, publication source (PeerJ) and either DOI or URL of the article must be cited.
License URL: https://creativecommons.org/licenses/by/4.0/

Keywords: Single-stage anaerobic digestion, Two-stage anaerobic digestion, High moisture fraction, Municipal solid waste, Biogas production, Organic loading, Incinerator fly ash, Nakhon si thammarat municipality, Microbial community, Methane production

Funding: Research and Development Institute Thaksin University (RDITSU) Thailand Research Fund through the Senior Research Scholar RTA6080010 Mid-Career Research Grant RSA6180048 This work was supported by the Research and Development Institute Thaksin University (RDITSU), Thailand Research Fund through the Senior Research Scholar (Grant No. RTA6080010) and Mid-Career Research Grant (Grant No. RSA6180048). The funders had no role in study design, data collection and analysis, decision to publish, or preparation of the manuscript.

==============================
Background

Anaerobic digestion (AD) is a suitable process for treating high moisture MSW with biogas and biofertilizer production. However, the low stability of AD performance and low methane production results from high moisture MSW due to the fast acidify of carbohydrate fermentation. The effects of organic loading and incineration fly ash addition as a pH adjustment on methane production from high moisture MSW in the single-stage AD and two-stage AD processes were investigated.

Results

Suitable initial organic loading of the single-stage AD process was 17 gVS L−1 at incineration fly ash (IFA) addition of 0.5% with methane yield of 287 mL CH4 g−1 VS. Suitable initial organic loading of the two-stage AD process was 43 gVS L−1 at IFA addition of 1% with hydrogen and methane yield of 47.4 ml H2 g−1 VS and 363 mL CH4 g−1 VS, respectively. The highest hydrogen and methane production of 8.7 m3 H2 ton−1 of high moisture MSW and 66.6 m3 CH4 ton−1 of high moisture MSW was achieved at organic loading of 43 gVS L−1 at IFA addition of 1% by two-stage AD process. Biogas production by the two-stage AD process enabled 18.5% higher energy recovery than single-stage AD. The 1% addition of IFA into high moisture MSW was useful for controlling pH of the two-stage AD process with enhanced biogas production between 87–92% when compared to without IFA addition. Electricity production and energy recovery from MSW using the coupled incineration with biogas production by two-stage AD process were 9,874 MJ ton−1 MSW and 89%, respectively.

Conclusions

The two-stage AD process with IFA addition for pH adjustment could improve biogas production from high moisture MSW, as well as reduce lag phase and enhance biodegradability efficiency. The coupled incineration process with biogas production using the two-stage AD process was suitable for the management of MSW with low area requirement, low greenhouse gas emissions, and high energy recovery.

Introduction

Municipal solid waste (MSW) has become a leading environmental concern due to its high quantity and the fact it contains a high amount of readily biodegradable organic waste. Landfills mostly treat MSW and require large areas, resulting in a lack of space for new landfills (Sukholthaman & Sharp, 2016). MSW incineration plays an increasingly important role in MSW management since it reduces the required area for new waste and efficiently reduces the volume of MSW (Yu et al., 2015). The incineration of MSW generates bottom ash, fly ash, and high moisture MSW as residue (Nie, 2008). The incineration of MSW generates bottom and fly ash in amounts of 250–300 and 25–50 kg ton−1, respectively (Jakob, Stucki & Kuhn, 1995). Fly ash is mainly composed of Si, Ca, Al, and Mg (Yu et al., 2015). The incineration of MSW is suitable for low moisture MSW, but high moisture MSW can remain untreated due to low calorific value. The leftover high moisture MSW creates an unpleasant odor that creates a severe environmental problem for the community around the incineration plant. Anaerobic digestion (AD) is a suitable biological process for treating high moisture MSW with biogas production and a semi-solid digestate as a fertilizer (Abdeshahian et al., 2016). Borowski (2015) reported that biogas yield from the co-digestion of MSW with sewage sludge reached 0.309 to 0.315 m3 kg−1VS under mesophilic conditions. However, the AD process indicates some limitations for the organic fraction of MSW with low stability, fast acidification due to high carbohydrate content, low methane production rate, and low VS degradation efficiency (Pavi et al., 2017).

The single-stage AD process is commonly applied for biogas production from high moisture MSW and faced with high volatile fatty acids accumulation and inhibition. A previous report from Michele et al. (2015) found that biogas production from the organic fraction of MSW causes reactor instability performance, as indicated by H2 concentrations of 8% in biogas and low pH (6.5). The low methane yield of 180 mL g−1 VS is observed in the AD of organic fraction of MSW containing high amounts of food waste (Forster-Carneiro et al., 2007). The instability of the AD systems feeding with the organic fraction of MSW is mostly influenced by VFAs accumulation, which inhibits methanogenic activity (Pavi et al., 2017). Adding ash to organic waste before being fed into the AD process improves the stability and reduces VFA accumulation. Ash releases alkali metals that contribute to increasing pH and buffer capacity in AD systems (Lo, 2005). The ash addition can control the pH of AD systems fed with the organic fraction of MSW at a suitable pH range (7.0–8.5) with enhanced VS degradation and methane production (Banks & Lo, 2003). The metals in the ash could act as co-enzymes during the AD process and enhance microbial growth. The improvement of biogas production by ash addition was also reported by Mamimin et al. (2019), who found that the addition of 5% oil palm fiber ash into palm oil mill effluent could improve biohythane production using the two-stage AD process. The two-stage AD process using the separation of acidogenic bacteria and methanogenic archaea enhance biogas production, substrate degradation efficiency, and the stability of the AD process (Demirel & Yenigün, 2002). The two-stage AD process increases methane production from olive mill waste by 10% when compared with the single-stage AD process (Rincón et al., 2009). The two-stage AD process has a higher reactor operation stability at high organic loading rates than the single-stage AD process. However, there are still few commercial applications for a two-stage AD digester. Therefore, this work aimed to evaluate methane production from high moisture MSW in both single-stage AD and two-stage AD processes. The effects of organic loading and incineration fly ash addition as a pH adjustment on biogas production of single-stage AD and two-stage AD processes were investigated.

Materials & Methods

Substrates and Inoculum

MSW was collected from landfill disposal sites located in Nakhon Si Thammarat Province, Thailand, from December 16 to 25, 2016. Roughly 200 kg of MSW was separated as low moisture MSW (<60%) and high moisture MSW (>60%) using the quartering method (Armijo de Vega, Ojeda Benítez & Ramírez Barreto, 2008). Incineration fly ash (IFA) was collected from the municipal solid waste incinerator at PJT Technology Co., Ltd., Phuket Province, Thailand. The high moisture MSW and IFA were dried at 95 °C until the moisture content was less than 10% (w/w). Dry MSW was milled with a hammer mill to 5 mm before being used as a substrate in the AD process. High moisture MSW was analyzed for pH, total solids (TS), volatile solids (VS), ash, protein, carbohydrates, and lipids according to APHA (2012). The IFA was analyzed for MgO, Al2O3, SiO2, K2O, CaO, TiO2, Fe2O3,Rb, SrO, Cl, Na2O, P2O5 and SO3content using an X-ray fluorescence spectrometer (Tan et al., 2002). The theoretical methane yield of high moisture MSW was calculated based on a modified Buswell Eqs. (1) and (2) from the CHON elemental composition (Buswell & Mueller, 1952). Hydrogen-producing sludge for the first-stage was collected from a hydrogen production reactor feeding with palm oil mill effluent. The sludge was cultivated on a 2 g L−1 sucrose medium for enhancing hydrogen-producing bacteria (O-Thong, Prasertsan & Birkeland, 2009). The enriched sludge with volatile suspended solids of 6.0 g L−1 was used as inoculum for the first stage (Mamimin et al., 2015). Methane production sludge was collected from biogas digester feeding with palm oil mill effluent. The anaerobic sludge was incubated at 35 °C for 10 days to remove the remaining organic materials. The sludge with volatile solids (VS) of 80 g L−1 was used as inoculum for second-stage and single-stage methane production (O-Thong et al., 2016). The composition of hydrogen and methane inoculum is shown in Table 1. (1) CaHbOcNd+4a−b−2c+3d4H2O→4a+b−2c−3d8CH4+a−4a+b−2c−3d8CO2+dNH3

(2) C29H48O14N+11H2O→17CH4+12CO2+NH3

Table 1 Characteristics of inoculum for hydrogen and methane production.

Parameters	Hydrogen producing sludge	Methane producing sludge	
Total solids (%)	2.99	9.72	
Total volatile solids (%)	2.38	8.37	
pH	5.52	7.83	
Alkalinity (mg-CaCO3/L)	2,400	5,200	

Biogas production from high moisture MSW by single-stage AD

Biogas production from high moisture MSW using single-stage AD was investigated at a working volume of 200 mL in a 1 L reactor under mesophilic conditions (Fig. 1). The high moisture MSW at initial VS loading of 9, 17, 26, 35, and 43 g-VS L−1 was mixed with methane-producing inoculum at a substrate to inoculum ratio (S:I) of 2:1 based on VS basis (Angelidaki et al., 2009). The 0.5% and 1% IFA were added into the mixtures as initial pH adjustment to 7.2–7.5. The mixtures were purged with nitrogen gas at a flow rate of 500 mL min−1 for 3 min to remove the oxygen in the reactor headspace. The reactors were closed with a rubber stopper and incubated under mesophilic conditions (35 °C) for 45 days. All treatments were done in triplicate. The biogas volume and gas composition were monitored daily using the water displacement method and gas chromatography, respectively.

Figure 1 Schematic diagram of the one-stage AD process (A) and two-stage AD process (B) for biogas production from high moisture MSW.

Biogas production from high moisture MSW by two-stage AD

The biogas production from high moisture MSW using the two-stage AD process was assayed as previously described by Mamimin et al. (2016). The reactor size of 1 L with a working volume of 200 mL was used for the first stage and 600 mL was used for the second stage (Fig. 1). Different initial VS loading of high moisture MSW at 9, 17, 26, 35, and 43 g-VS L−1 was mixed with hydrogen-producing sludge at S:I of 20:1 based on VS basis for the first stage (Mamimin et al., 2019). The 0.5 and 1.0% IFA was added into the mixtures for initial pH adjustment to 5.5-6. The mixture was added to the reactors and closed with a rubber stopper. The reactors were incubated under mesophilic conditions (35 °C) for 15 days. After 15 days, the reactors were opened in a nitrogen environment and introduced methane-producing inoculum at S:I ratio of 2:1 based on the VS basis for the second stage AD process. The reactors were closed with a rubber stopper and continue incubated at mesophilic conditions (35 °C) for 45 days. All of the treatments were done in triplicate. The volume of biogas was measured by the water displacement method. The gas composition was determined by gas chromatography. The biogas was taken once every day in the first stage for hydrogen gas analysis. The biogas was taken from the second stage every day in the first week and then every 2 days thereafter for methane gas analysis. The microbial community responsible for two-stage AD was analyzed by polymerase chain reaction-denaturing gradient gel electrophoresis (PCR-DGGE) techniques.

Microbial community analysis

Sludge samples from an optimum condition for biogas production by two-stage AD processes were taken for PCR-DGGE analysis. The genomic DNA of sludge samples was extracted using a method previously described by Kuske et al. (1998). The DNA quality was checked by agarose gel electrophoresis and used as a template for the two-step polymerase chain reaction (PCR). The first PCR for the 16S rRNA of the archaea population was amplified by primer Arch21f (5′ TTCCGGTTGATCCYGCCGGA 3′) and Arch958r (5′ YCCGGCGTTGAMTCCAATT 3′). The first PCR for the 16S rRNA of the bacteria population was amplified by primer 1492r (5′ GAAAGGAGGTGATCCAGCC 3′) and 27f (5′ GAGTTTGATCCTTGGCTCAG 3′). PCR amplification was conducted in an automated thermal cycler with pre-denaturation at 95 °C for 5 min followed by 25 cycles of denaturation at 95 °C for 30 s, annealing at 52 °C for 40 s, elongation at 72 °C for 90 s, and post-elongation at 72 °C for 5 min. The reactions were subsequently cooled to 4 °C (O-Thong, Prasertsan & Birkeland, 2009). The second PCR was amplified from the amplicons of the first PCR as a DNA template with primer Arch519r (5′ TTACCGCGGCKGCTG 3′ with 40 bp GC clamp) and Arch340f (5′ CCTACGGGGYGCASCAG 3′) for archaea population. The second PCR of bacteria population was amplified with primer 518r (5′ ATTACCGAGCTGCTGG 3′ with 40 bp GC-clamp) and 357f (5′ CCTACGGGAGGCAGCAG 3′). PCR products were analyzed on agarose gel electrophoresis before DGGE analysis. The amplicons from the second PCR were used for DGGE analysis, as previously described by Prasertsan, O-Thong & Birkeland (2009). The DGGE bands were excised from the gel and re-amplified under similar conditions as the second PCR. The PCR product was purified and sequenced by Macrogen Inc. (Seoul, Korea). The identification of 16S rRNA gene sequences from DGGE bands was carried out by BLAST database searches in GenBank (Tatusova et al., 2016).

Analytical methods

The composition of high moisture MSW was analyzed for pH, TS, VS, protein, carbohydrates, and lipids according to standard methods (APHA, 2012). The chemical composition of high moisture MSW with regards to the elements C, H, O, and N was analyzed according to Lesteur et al. (2010). The total energy of MSW was analyzed using a bomb calorimeter (GE-5055 Compensated Jacket Calorimeter, Parr, Illinois, USA). Biogas compositions were analyzed by gas chromatography (GC-8A Shimadzu, Kyoto, Japan) equipped with thermal conductivity detectors (TCD) and fitted with a 2.0 m packed column (Shin-Carbon ST 100/120 Restek). Volatile fatty acids (VFA) were analyzed by gas chromatography (GC-17A, Shimadzu, Kyoto, Japan) equipped with a flame ionization detector (FID) and Stabilwax-DA column (dimensions 30 m × 0.32 mm × 0.25 mm). The cumulative methane yield from a single-stage and two-stage AD process was fitted for hydrolysis rate constant (kh), as in Eq. (3) below. (3) lnB∞−BB∞=−kht

where B∞ is the ultimate biogas yield, and B is the biogas yield at a given time (t). The digestion kinetics of high moisture MSW in the single-stage and two-stage AD system were evaluated by the modified Gompertz equation. The lag phase (d) and biogas production rate (mL gVS−1 d−1) were also estimated from the modified Gompertz equation. (4) Bt=Pmax× exp−expRmax×ePmax×λ−t+1,t≥0

where B(t) is the specific hydrogen and methane yield of high moisture MSW at a given time (mL g−1VS). Pmax is the maximum hydrogen and methane potential (mL g−1VS). t is the digestion time (d). R is the maximum hydrogen and methane production rate (mL g−1VS d−1). λ is the lag-times. e is the exponential of 1, which is 2.71828 (Zhen et al., 2016).

Results

High moisture MSW and IFA composition

High moisture MSW from a landfill site in Nakhon Si Thammarat, Thailand was mainly composed of food waste, green waste, fruit waste, and vegetable waste. The MSW contained 53.6% high moisture waste and 46.4% low moisture waste. High moisture MSW had TS, VS, ash, and moisture of 26.36%, 18.34%, 8.02%, and 65.62%, respectively (Table 2). The high moisture MSW contained high carbohydrate and protein content. The protein, carbohydrates, lipids in the high moisture MSW amounted to 20.4%, 38.6%, and 11.0% of TS, respectively. Incineration fly ash (IFA) was mainly composed of CaO, Cl, Na2O, K2O, and SO3 at 39.6, 22.1, 8.35, 4.95, and 3.22% of TS, respectively (Table 3).

Table 2 Characteristics of high moisture municipal solid waste from Nakhon Si Thammarat landfill site, Thailand.

Parameters	High moisture MSW	
pH	5.6	
Total solids (%w/w)	26.4	
Volatile solids (%w/w)	18.3	
Ash (%w/w)	8.02	
Moisture (%w/w)	73.6	
C (%)	51.2	
H (%)	66.6	
O (%)	40.3	
N (%)	2.0	
S (%)	0.1	
Protein (% of TS)	20.4	
Carbohydrate (% of TS)	38.6	
Lipid (% of TS)	11.0	

Table 3 Composition of incineration fly ash from incineration of high moisture municipal solid waste.

Elements	Value (% of TS)	
Na2O	8.35	
MgO	1.17	
Al2O	0.663	
SiO2	1.65	
P2O5	0.689	
SO3	3.22	
Cl	22.1	
K2O	4.95	
CaO	39.6	
TiO2	0.367	
Cr2O3	0.014	
MnO	0.023	
Fe2O3	0.432	
NiO	0.007	
CuO	0.055	
ZnO	0.427	
Br	0.108	
Rb2O	0.026	
SrO	0.050	
CdO	0.021	
SnO2	0.048	
Sb2O3	0.026	
PbO	0.103	

Biogas production from high moisture MSW by the single-stage AD process

The theoretical methane yield of high moisture MSW was 576 mL g−1 VS, while real biogas production using the single-stage AD process was 268-287 mL-CH4 g−1 VS. A maximum methane yield of 287 mL-CH4 g−1 VS was achieved at an initial VS loading of 17 g-VS L−1 with IFA addition of 0.5%(w/v). The methane yields at initial VS loading of 9, 17, 26, 35, 43 gVS L−1 with 0.5%(w/v) IFA addition were 220, 287, 179, 10.5, and 5.08 mL CH4 g−1VS, respectively (Fig. 2). The methane yields at initial VS loading of 9, 17, 26, 35, 43 gVS L−1 with 1%(w/v) IFA addition were 238, 268, 218, 16.4, and 5.35 mL CH4 g−1 VS, respectively. The results indicated that methane yield at initial VS loading of 9-17 gVS L−1 at both IFA addition was significantly (p < 0.05) higher than the initial VS loading of >17 gVS L−1. The single-stage AD process could completely degrade high moisture MSW at initial loading of 9-26 gVS L−1 at both IFA addition. The VFAs concentration of the single-stage AD process at initial VS loading of 9-26 gVS L−1 at IFA addition of 0.5% and 1%(w/v) were 178–267 and 186–240 mg L−1, respectively (Table 4). The initial VS loading of >26 gVS L−1 had low methane yield (5.08–16.4 mL-CH4 g−1 VS) and high VFA accumulation (1.2–2.2 g L−1) at both IFA addition. The initial VS loading of >26 gVS L−1 had high acetic acid, propionic acid, and butyric acid in the AD system. The acetic acid, propionic acid, and butyric acid concentration in the AD system at initial VS loading of >26 gVS L−1 were 553-979, 105-187, 541-1026 mg L−1, respectively (Table 5). The total alkalinity of the single-stage AD process at all initial VS loading at IFA addition of 0.5 and 1%(w/v) was 2,600-4,600 and 2,450–3,500 mg-CaCO3 L−1, respectively. The initial VS loading of >26 gVS L−1 at both IFA addition had a VFA to alkalinity ratio higher than 0.3, indicating the imbalance of the AD process for biogas production. High moisture MSW without IFA addition had no methane production due to high VFAs accumulation (1,235 mg L−1) and low alkalinity (508 mg-CaCO3 L−1), leading to low pH and an inhibited AD process. The high methane production rate of 14.2–15.3 was also achieved at an initial VS loading of 17 gVS L−1. The lag phase of the single-stage AD process was 6-11 days at an initial VS loading of 17 gVS L−1; increasing the VS loading extended the lag phase. Suitable initial organic loading of the single-stage AD process was 17 gVS L−1 at IFA addition of 0.5% with methane yield of 287 mL CH4 g−1 VS. Maximum methane production of 48.7 m3 CH4 ton−1 high moisture MSW was achieved at initial VS loading of 17 gVS L−1 at IFA addition of 0.5%. Maximum biodegradation efficiency of 50% and 47% was achieved at initial VS loading of 17 gVS L−1 with IFA addition of 0.5 and 1%(w/v), respectively. Methane production, methane yield, and biodegradation efficiency of high moisture MSW in the single-stage AD process at IFA addition of 0.5% (w/v) were not significantly different (p > 0.05) with IFA addition of 1% (w/v). The methane production and methane yield of high moisture MSW with IFA addition were significantly (p < 0.05) higher than without IFA addition.

Figure 2 Cumulative methane yield from single-stage anaerobic digestion of high moisture municipal solid waste with 0.5% (A) and 1.0% (B) addition of IFA for pH adjustment.

Table 4 Process performance of single-stage anaerobic digestion of high moisture MSW.

Initial VS loading (g-VS L−1)	IFA addition (%w/v)	Methane yield (mL CH4 g−1 VS)	Methane production (m3 CH4 ton−1 MSW)	Methane production rate (mL CH4 g−1 VS d−1)	Lag phase (d)	Hydrolysis constant (d−1)	VFAs (mg L−1)	Alkalinity (mg-CaCo3 L−1)	Biodegradation (%)	
9	0.5	220	37.5	18.4	6.07	0.137	209	4,600	38.3	
17	0.5	287	48.7	15.3	10.6	0.108	198	2,600	49.8	
26	0.5	179	30.4	6.96	18.8	0.058	178	3,225	31.1	
35	0.5	10.5	1.78	0.34	–	0.063	1,218	2,925	1.82	
43	0.5	5.08	0.86	0.78	–	0.089	1,267	3,050	0.88	
9	1	238	40.4	18.9	6.66	0.128	193	2,450	41.2	
17	1	268	45.6	14.2	11.7	0.112	224	2,450	46.5	
26	1	218	37.1	14.1	30.1	0.043	205	2,875	37.9	
35	1	16.4	2.78	5.88	–	0.038	2,240	2,950	2.84	
43	1	5.35	0.91	0.69	–	0.081	2,186	3,500	0.93	
9	0	0	0	0	0	0	1,235	508	0.30	

Table 5 Volatile fatty acids distribution in single-stage anaerobic digestion effluent of high moisture MSW.

Initial VS loading (g-VS L−1)	IFA addition (%w/v)	Acetic acid (mg L−1)	Propionic acid (mg L−1)	Isobutyric acid (mg L−1)	Butyric acid (mg L−1)	Isovaleric acid (mg L−1)	Valeric acid (mg L−1)	TVFAs (mg L−1)	
9	0.5	88.2	13.9	4.4	99.1	3.4	0	209	
17	0.5	85.4	13.4	3.4	92.9	3.0	0	198	
26	0.5	76.1	11.6	2.9	84.8	2.7	0	178	
35	0.5	531.8	78.1	18.7	571.3	18.0	0	1,218	
43	0.5	535.5	115.3	22.6	574.3	19.2	0	1,267	
9	1	85.3	11.8	3.0	89.8	3.1	0	193	
17	1	97.3	13.8	3.8	105.3	3.7	0	224	
26	1	89.7	11.8	3.1	97.3	3.0	0	205	
35	1	972.2	143.5	35.0	1028.1	38.1	23.1	2,240	
43	1	979.5	187.2	30.7	958.6	30.1	0	2,186	
9	0	553.4	105.7	17.3	541.6	17.0	0	1,235	

Hydrogen and methane production from high moisture MSW by the two-stage AD process

Biogas production and process performance of high moisture MSW using the two-stage AD process at different initial VS loading and IFA addition for pH adjustment are shown in Table 6. Between 80–90% of hydrogen was produced within 4-6 days in all experiments with IFA addition. Hydrogen content in the biogas ranged from 30–40%. The lag phase for hydrogen production of high moisture MSW in the first stage was 0.4–1.0 d. The hydrolysis rate of high moisture MSW in the first stage was 0.22–0.87 d−1. The biodegradation of high moisture MSW in the first stage AD process was 9–10%. The hydrogen yields of first stage AD at initial VS loading of 9, 17, 26, 35, 43 gVS L−1 with 0.5% IFA addition were 40.8, 47.6, 46.4, 43.3, and 42.5 mL H2 g−1 VS, respectively (Fig. 3A). The hydrogen yields at initial VS loading of 9, 17, 26, 35, 43 gVS L−1 with 1% IFA addition were 16.4, 41.3, 43.0, 36.1, and 47.4 mL H2 g−1VS, respectively (Fig. 3C). The hydrogen yield of high moisture MSW without IFA addition was 8.05 mL H2 g−1 VS with a VFA concentration of 3,328 mg L−1. The IFA addition (0.5–1%w/v) increased hydrogen yield 2-6 times compared to high moisture MSW alone. The hydrogen yields of all initial VS loading at both IFA addition were similar, while the hydrogen production rate at low initial VS loading of 9-17 gVS L−1 (7-16.2 mL H2 g−1VS d−1) was significantly (p < 0.05) higher than high initial VS loading of 26–43 gVS L−1 (8–11 mL H2 g−1VS d−1) (Table 6). The high VFAs concentration of 1,185-2,066 mg L−1 was observed in hydrogen effluent at all VS loading with 0.5 and 1%(w/v) IFA addition. The total alkalinity of high moisture MSW with IFA addition ranged from 900–1,300 mg-CaCO3 L−1, while the high moisture MSW without IFA addition had low total alkalinity of 315 mg-CaCO3 L−1. Low buffer capacity was observed in the AD of high moisture MSW without IFA addition. The addition of ash into high moisture MSW supports buffer capacity and prevents inhibition from low pH. The maximum hydrogen production of high moisture MSW with IFA addition was 8.7 m3 H2 tons−1 of high moisture MSW, while the hydrogen production of high moisture MSW without IFA addition was 1.39 m3 H2 tons−1of high moisture MSW. The IFA addition (0.5–1%w/v) improves hydrogen production from high moisture MSW by 82% when compared with hydrogen production from high moisture MSW without IFA addition.

Table 6 Process performance of two-stage anaerobic digestion of high moisture municipal solid waste (MSW) from Nakhon Si Thammarat landfill site, Thailand.

Hydrogen production stage	
Initial VS loading (g-VS L−1)	IFA addition (%)	Hydrogen yield (mL H2 g−1VS)	Hydrogen production (m3 H2 tonne−1 MSW)	Hydrogen production rate (mL H2 g−1VS d−1)	Lag phase (d)	Hydrolysis constant (d−1)	VFAs (mg L−1)	Alkalinity (mg-CaCO3 L−1)	Biodegradation (%)	
9	0.5	40.8	6.94	16.20	0.48	0.871	1,607	1,050	9.1	
17	0.5	47.6	8.10	12.70	0.79	0.502	1,185	900	10.6	
26	0.5	46.4	7.90	8.58	0.78	0.412	1,706	925	10.3	
35	0.5	43.3	7.36	6.33	0.14	0.301	1,771	1,087	9.6	
43	0.5	42.5	7.23	5.65	0.14	0.229	2,066	1,062	9.4	
9	1	16.4	2.80	7.96	1.02	0.331	1,350	950	3.6	
17	1	41.3	7.03	15.88	1.08	0.522	1,195	987	9.2	
26	1	43.0	7.32	11.03	0.81	0.504	1,423	1,050	9.6	
35	1	36.1	6.14	8.45	0.47	0.463	2,020	1,300	8.0	
43	1	47.4	8.06	8.11	0.24	0.358	1,680	1,075	10.5	
9	0	8.05	1.39	1.61	0.68	0.348	3,328	315	1.8	
Methane production stage	
Initial VS loading (g-VS L−1)	IFA addition (%)	Methane yield (mL CH4 g−1VS)	Methane production (m3 CH4 tonne−1 MSW)	Methane production rate (mL CH4 g−1VS d−1)	Lag phase (d)	Hydrolysis constant (d−1)	VFAs (mg L−1)	Alkalinity (mg-CaCO3 L−1)	Biodegradation (%)	
9	0.5	399	67.9	46.20	1.24	0.133	276	3,050	69.3	
17	0.5	396	67.4	47.30	1.61	0.129	152	3,500	68.9	
26	0.5	400	68.1	45.10	1.86	0.113	171	3,450	69.6	
35	0.5	362	61.6	36.40	1.83	0.106	183	3,600	62.9	
43	0.5	319	54.2	30.70	1.85	0.101	78	3,175	55.5	
9	1	348	59.3	36.70	0.85	0.106	162	3,075	60.5	
17	1	369	62.2	41.40	1.37	0.121	197	3,400	64.1	
26	1	375	63.8	42.90	1.71	0.114	184	3,525	65.1	
35	1	341	58.1	33.60	1.61	0.103	207	3,700	59.3	
43	1	363	61.8	33.70	1.91	0.103	176	3,300	63.1	
9	0	315	52.8	27.73	0.952	0.104	247	3,100	54.8	

Figure 3 Cumulative hydrogen and methane yield from two-stage anaerobic digestion of high moisture municipal solid waste with incineration fly ash addition for pH adjustment at 0.5% (A–B) and 1% (C–D).

The homogenized hydrogen effluent from the first stage was used as a substrate for methane production in the second stage. The methane yields of the second stage at initial VS loading of 9, 17, 26, 35, 43 gVS L−1 with 0.5% IFA addition were 399, 396, 400, 362, and 319 mL CH4 g−1VS, respectively (Fig. 3B). The methane yields at initial VS loading of 9, 17, 26, 35, and 43 gVS L−1 with 1% IFA addition were 348, 369, 375, 341, and 363 mL CH4 g−1VS, respectively (Fig. 3D). Methane yield of high moisture MSW without IFA addition in the second stage was 315 mL CH4 g−1VS. The IFA addition (0.5 and 1%w/v) improved methane yield 15–20% from high moisture MSW via the two-stage AD process. The lag phase of methane production in the two-stage AD process (0.85–1.91 days) was shorter than the single-stage AD process (6-30 days) (Table 6). IFA addition to high moisture MSW effectively increased biogas production in the second stage. Stable alkalinity, high biodegradation efficiency, low VFAs accumulation, and higher methane production were achieved in the two-stage AD process. The VFAs and total alkalinity after methane production ranged between 78-276 mg L−1 and 3,050-3,700 mg-CaCO3 L−1, respectively (Table 7). Suitable initial organic loading of the two-stage AD process was 43 gVS L−1 at IFA addition of 1% with methane yield of 363 mL CH4 g−1 VS. Maximum methane production of 66.6 m3 CH4 tons−1 of high moisture MSW was achieved from high moisture MSW at the initial VS loading of 43 gVS L−1 with 1.0% IFA addition corresponded to maximum biodegradation efficiency of 63.1%. The highest hydrogen and methane production of 8.7 m3 H2 ton−1 high moisture MSW and 66.6 m3 CH4 ton−1 high moisture MSW was achieved at organic loading of 43 gVS L−1 at IFA addition of 1% by the two-stage AD process. Biogas production by the two-stage AD process showed 18.5% higher energy recovery than the single-stage AD process. The addition of IFA at 1% into high moisture MSW was useful for controlling pH for the two-stage AD process with enhanced biogas production between 87–92% when compared to without IFA addition.

Table 7 Volatile fatty acids profile of two-stage anaerobic digestion of high moisture municipal solid waste (MSW) from Nakhon Si Thammarat landfill site, Thailand.

Hydrogen production stage	
Initial VS loading (g-VS L−1)	IFA addition (%w/v)	Acetic acid (mg L−1)	Propionic acid (mg L−1)	Isobutyric acid (mg L−1)	Butyric acid (mg L−1)	Isovaleric acid (mg L−1)	Valeric acid (mg L−1)	TVFAs (mg L−1)	
9	0.5	678.1	156.2	24.4	721.5	26.9	0.0	1,607	
17	0.5	454.0	153.2	0.0	559.4	18.4	0.0	1,185	
26	0.5	766.8	102.0	25.0	786.6	25.6	0.0	1,706	
35	0.5	732.1	239.0	0.0	776.6	23.3	0.0	1,771	
43	0.5	805.8	336.4	29.3	866.1	28.4	0.0	2,066	
9	1	508.7	161.1	20.0	635.7	24.5	0.0	1,350	
17	1	493.6	157.6	0.0	522.2	21.6	0.0	1,195	
26	1	503.7	242.0	0.0	657.4	19.9	0.0	1,423	
35	1	689.1	377.6	37.1	863.3	25.8	27.2	2,020	
43	1	682.3	267.8	19.7	685.9	24.3	0.0	1,680	
9	0	1351.5	530.5	39.1	1358.7	48.1	0.0	3,328	
Methane production stage	
Initial VS loading (g-VS L−1)	IFA addition (%w/v)	Acetic acid (mg L−1)	Propionic acid (mg L−1)	Isobutyric acid (mg L−1)	Butyric acid (mg L−1)	Isovaleric acid (mg L−1)	Valeric acid (mg L−1)	TVFAs (mg L−1)	
9	0.5	88.95	12.80	3.12	88.37	82.47	0.00	276	
17	0.5	66.82	9.71	2.25	71.30	2.38	0.00	152	
26	0.5	76.66	10.20	2.50	78.64	2.56	0.00	171	
35	0.5	82.04	10.81	2.43	85.34	2.84	0.00	183	
43	0.5	64.42	7.93	0.00	5.77	0.18	0.00	78	
9	1	72.27	9.64	2.40	75.62	2.40	0.00	162	
17	1	83.72	12.67	3.11	94.44	3.21	0.00	197	
26	1	79.26	11.79	2.54	87.60	2.93	0.00	184	
35	1	91.29	12.32	3.11	97.38	3.10	0.00	207	
43	1	79.41	11.20	2.43	80.11	2.55	0.00	176	
9	0	108.82	14.68	3.71	116.09	3.70	0.00	247	

Microbial community of the two-stage AD process

Bacteria and archaea community structures of the two-stage AD process at initial VS loading of 43 gVS L−1 with the IFA addition of 1% are shown in Fig. 4. The bacterial community in the first stage for hydrogen production was composed of Clostridium sp., Sphingobacterium sp., Gramella sp., Eubacterium sp., and Lactobacillus sp. No archaea were found in the first stage. Clostridium sp. and Sphingobacterium sp. were dominated in the first stage and involved in hydrogen production from high moisture MSW. The bacterial community in the second stage for methane production was composed of Clostridium sp., Sulfurihydrogenibium sp., Gramella sp., Lutaonella sp., Sphingobacterium sp., Cellulophaga sp., and Flavobacterium sp. The archaeal community of the second-stage was dominated by hydrogenotrophic and acetoclastic methanogen. The archaea community of the second stage was composed of Methanobacterium sp., Methanocaldocccus sp., and Methanothermus sp. The two-stage AD process was dominated by Clostridium sp., Sphingobacterium sp., Methanobacterium sp., and Methanothermus sp., which were responsible for hydrogen and methane production.

Figure 4 Bacterial community in the first stage (S1-BACT), bacterial community in the second stage (S2-BACT), and archaea community in the second stage (S2-ARCH) of the two-stage anaerobic digestion process of high moisture municipal solid waste for hydrogen and methane production at an initial volatile solids loading of 43 g-VS L-1 and IFA addition at 1.0%.

Energy recovery

Energy recovery from MSW using the coupled incineration process with biogas production by the two-stage AD process is shown in Fig. 5. Municipal solid waste management by landfills has no energy recovery and high greenhouse gas emissions (1,360 kg CO2-eq ton−1 MSW). The greenhouse gas emissions from MSW management by landfills comprise mainly methane emissions (64.65 kg CH4 ton−1), which corresponds to 1,360 kg CO2-eq of GHG emissions. The management of MSW by incineration can recover energy from low moisture MSW at 7,231 MJ ton−1. The remaining high moisture MSW accounts for 54% with no energy recovery and high greenhouse gas emissions (762 kg CO2-eq ton−1 MSW). The management of MSW by incineration is better than landfilling in terms of energy recovery and greenhouse gas emissions. Energy recovery from MSW via the incineration process remains at 66%. The coupled incineration process with biogas production using the two-stage AD process for MSW management can significantly increase energy recovery by up to 89%. Energy production from high moisture MSW in the form of biogas via a two-stage AD process with IFA addition was 2,553 MJ ton−1. Energy production from the coupled incineration process with biogas production using the two-stage AD process for management of MSW was 9,784 MJ ton−1. The coupled incineration process with biogas production using the two-stage AD process for management of both low moisture and high moisture MSW could reduce landfill area as well as the emission of greenhouse gases (GHG). High moisture MSW treated via the two-stage AD process could decrease GHG emissions by 762 kg CO2-eq ton−1 MSW. Therefore, the coupled incineration process with biogas production using the two-stage AD process could provide a solution for reducing GHG and boost efforts to achieve sustainable development for MSW management.

Figure 5 Energy recovery from municipal solid waste by the coupled landfill and incineration process (A) and coupled incineration process and two-stage anaerobic digestion (B).

Discussion

High biogas production from high moisture MSW with IFA addition by the single-stage AD process was achieved at low VS loading (9–17 gVS L−1) with low VFAs accumulation (<300 mg L−1). The excellent AD performance of organic fraction MSW was obtained at low VS loading (Yan et al., 2019). Mattioli et al. (2017) also found that optimum VS loading of organic fraction MSW was 29 gVS L−1 with maximum methane yield of 270 ml-CH4 g−1 VS by a single-stage AD reactor. The alkalinity was in line with previously reported 3,000-5,000 mg-CaCO3 L−1 (Angelidaki et al., 2009). The VS loading of >26 gVS L−1 for both IFA addition (0.5% and 1%) had a VFA to alkalinity ratio higher than 0.3, indicating the imbalance of the AD process for biogas production (Khanal, 2008). The volatile fatty acids/alkalinity ratio should be maintained at 0.10–0.30 to avoid acidification of the AD process (Barampouti, Mai & Vlyssides, 2005). The low buffered and fast acidified high moisture MSW resulted in an imbalance of the single-stage AD process due to the quick change of pH under the high VS loading (Zhang, Qiu & Chen, 2012). High moisture MSW with IFA addition for pH adjustment could improve the self-buffering capacity to meet the demands of microbial growth (Zhang et al., 2016). The results were confirmed by Podmirseg et al. (2013), who found that the loading of 0.5 g ash g−1 TS could enhance biogas production as well as improve the hydrolysis rate. The IFA addition of 0.5–1%w/v into high moisture MSW could improve hydrogen yield (2–6 times) and methane yield (0.2–0.5 times) for the two-stage AD process. High moisture MSW contains high carbohydrates (including rice), making it a suitable substrate for hydrogen production and the immediate generation of hydrogen after inoculation (Dong et al., 2009). Mamimin et al. (2019) reported that ash addition into palm oil mill effluent enhanced hydrogen production and hydrogen yield by the two-stage AD process. Microelements in ash are vital for the enzymes involved in the biological hydrogen production pathway, resulting in the high degradation efficiency of substrates and high hydrogen yield (Mamimin et al., 2019; Thanh et al., 2016). The trace metals in IFA were possibly metabolized as micronutrients in the first stage of hydrolytic and acidogenic bacteria (Lo, 2005).

Hydrolytic and acidogenic bacteria were dominated in the first stage. Clostridium sp., Sphingobacterium sp., Eubacterium sp., and Lactobacillus sp. are very useful in the degradation of lipids, carbohydrates, and proteins (Martín-González et al., 2011). The main compositions of high moisture MSW were carbohydrates, proteins, and lipids. Yuan et al. (2012) found that Clostridium sp. could utilize various carbon sources such as cellobiose, glucose, xylose, and sucrose with a volatile fatty acid, carbon dioxide, and hydrogen production. The archaeal community of the second-stage was dominated by hydrogenotrophic and acetoclastic methanogen. Methanobacterium sp. can utilize H2/CO2 and formate as substrates for methane production (Yang et al., 2015). Luo et al. (2015) reported that biochar could enrich Methanobacterium sp. when added to the AD system. Methanobacterium sp. was dominated in the AD system with biochar addition. IFA addition into high moisture MSW enhanced biogas production, the diversity of bacteria, and the diversity of archaea by acting as co-enzymes and buffer capacity during the AD process. The populations of Methanosaeta sp., Methanobacteriales, Methanobacterium sp., Methanococcales were increased by 208%, 133%, 50%, and 144%, respectively, after proper pH adjustment of the AD systems (Zahedi et al., 2016).

Conclusions

The two-stage AD process enhances methane production and biodegradation efficiency with a short lag phase from high moisture MSW. Hydrogen and methane production of 7.9 m3 H2 ton−1 high moisture MSW and 68.1 m3 CH4 ton−1 high moisture MSW, respectively, was achieved at an initial loading of 26 gVS L−1 and 1% IFA addition. The IFA addition has excellent potential for control during digested high moisture MSW using a two-stage AD process with 87–92% improvement of biogas production compared to without ash addition. The biogas production from high moisture MSW by two-stage AD has 18.5% higher energy recovery than a single-stage AD. The coupled incineration with a two-stage biogas production for treating 1-ton of MSW has electricity production of 9,874 MJ with an energy recovery of 89%. Coupled incineration with biogas production via the two-stage AD process is suitable for completely utilizing MSW with low land area requirement, low greenhouse gas emission, and high energy recovery.

Supplemental Information

Supplemental Information 1 Biogas production of single stage AD

Click here for additional data file.

Supplemental Information 2 Biogas production of two stage AD

Click here for additional data file.

Supplemental Information 3 Composition of municipal solid waste

Click here for additional data file.

The authors would like to thank Nakhon Si Thammarat Municipality and Phuket Municipality, Thailand, for providing the municipal solid waste and incineration fly ash.

Additional Information and Declarations

Competing Interests

Author Contributions

Field Study Permissions

Data Availability

The authors declare there are no competing interests

Wattananarong Markphan and Wantanasak Suksong performed the experiments, analyzed the data, prepared figures and/or tables, authored or reviewed drafts of the paper, and approved the final draft.

Chonticha Mamimin and Poonsuk Prasertsan conceived and designed the experiments, authored or reviewed drafts of the paper, and approved the final draft.

Sompong O-Thong conceived and designed the experiments, analyzed the data, authored or reviewed drafts of the paper, and approved the final draft.

The following information was supplied relating to field study approvals (i.e., approving body and any reference numbers):

Nakhon Si Thammarat Municipality and Phuket Municipality, Thailand provided the samples of municipal solid waste and incineration fly ash.

The following information was supplied regarding data availability:

The raw measurements are available in the Supplemental Files.

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
