# Peer review of "Comparative assessment of single-stage and two-stage anaerobic digestion for biogas production from high moisture municipal solid waste"

_PeerJ, doi:10.7717/peerj.9693_

## Round 0.1 · original submission · Major Revisions

Besides the comments addressed by the reviewers, the manuscript requires careful language checking. Mistakes are found in the entire text. Several sentences must be re-written. See some examples:

Background: Municipal solid waste (MSW) managed by the incineration method generating high moisture MSW as residue (?)....

Results: Maximum methane yield from high moisture MSW by single-stage process and the two-stage process was 218 and 400 mL CH4 g-1 VS.(?)...

Conclusion: Biogas productionfrom high (?) MSW by two-stage AD process with IFA addition for pH adjustment could improve methane production, the lag phase of the AD process, and biodegradability efficiency when compared with the single-stage AD process.
You probably mean "Two-stage AD process with IFA addition for pH adjustment could improve the methane production from high moisture MSW, etc...

Reviewer 1 ·

Basic reporting

It contains what is necessary to be published.

Experimental design

On line 110 with respect to IFA 0.5% and 1%, is it related to the VS load? or even the pH adjustment? or by weight ?, explain.
In the results appears the aforementioned relationship line 184 (w / v)
Doesn't it explain what type of reactor it is, agitated tank ?, drained bed? in batch, etc. is very important.

Validity of the findings

In general it is a good article that shows the incorporation of elements to buffer pH, in addition to evaluating the microbial populations that are in each stage and explains their influence on the process.

Additional comments

On line 110 with respect to IFA 0.5% and 1%, is it related to the VS load? or even the pH adjustment? or by weight ?, explain.
In the results appears the aforementioned relationship line 184 (w / v)
Doesn't it explain what type of reactor it is, agitated tank ?, drained bed? in batch, etc.
in line 125 it mentions the alkalinity range in mg-CaCO3 and in the discussion it does not compare with the appropriate ones.

In general it is a good article that shows the incorporation of elements to buffer pH, in addition to evaluating the microbial populations that are in each stage and explains their influence on the process.

Reviewer 2 ·

Basic reporting

1. The authors should define the uncommon abbreviations at their first mention. For example, L27: VS, L31: AD.
2. The definition of abbreviation IFA was needed at first mention and authors defined it twice at lines 91 and 93. Same problem of the abbreviation VS (volatile solids) was found at line 95 and 103.
3. Lines 110-111, the sentence “The 0.5% and 1% of IFA were added for pH adjustment in all initial VS loading” could be improved- the current phrasing makes comprehension difficult.

Experimental design

1. The procedure of PCR should be provided. And more details or the references of DGGE should be added.

Validity of the findings

no comment

Additional comments

1. Lines 109 and 121, Why the initial VS loading of 9, 17, 26, 35, and 43 g-VS L-1 was mixed with different v/v ratio by single-stage AD (30%) and two-stage AD (20%)?
2. The substrate to inoculum ratio (S:I) of 20:1 was used at two-stage AD process and the S:I was 2:1 at single-stage AD process. The authors should provide more details or reference about the difference.
3. Lines 277-278, I suggest that authors should provide the references for the reason of the suitable VS loading was <26 gVS L-1.

---

## Round 0.2 · Minor Revisions

Your revised version does not show those parts changed, as required by the reviewers.

Language checking was not conducted properly, as requested. Many mistakes are still found in the revised text. Below, some examples:

26-29
However, anaerobic digestion of high moisture MSW easy to acidified by carbohydrate fermentation resulting in the low stability of AD performance, low methane production, and low volatile solids (VS) degradation efficiency.

99-100
The high moisture MSW was collected from disposal sites at Nakhon Si Thammarat 100 Municipality, Thailand (Fig. 1).
A description of the reactor used should be added instead of the image from the disposal site.

101-102
The 200 kg of MSW 102 was manually divided into high moisture MSW and low moisture MSW (Fig. 2).

102-103
An incineration fly ash (IFA)

104
IFA were dried at temperature 95°C until the moisture content less than 10% (w/w)

109
The characteristics of high moisture MSW and IFA was shown in

371
production from high moisture MSW, reduce lag phase, and enhanced biodegradability efficiency.

377-388
The electricity recovery via combine incineration with

380
for management both low moisture.

Other mistakes are found all over the text. Please, conduct a proper language checking.

Figure 1 does not add any relevant information to the paper and should be deleted. Instead, a description of the apparatus/reactor is missing.

Figure 2 is not necessary. Instead, the information should be better presented in a simple Table.

I suggest you send once more the manuscript after the above-mentioned corrections. This time, I kindly ask you to use tracking change or another color to highlight the changes you made.

Reviewer 3 ·

Basic reporting

The authors have focussed on the biogas production from high moisture containing MSW using single and two AD. However, the authors have not much focussed on the volatile fatty acids production during two stage AD. Quantifying the VFA production would be a value added product apart from biogas if two stage AD is used.
The authors may try to add VFA production profile if the data is available
The authors did not define "IFA" for the first time in abstract. So it is not clear what actually is IFA meant for...
In the abstract, AD has been defined, however it is not used as AD further. For instance, however, anaerobic digestion...... this should be AD
So, the authors may thoroughly check the grammatical errors, typos, and the sentence formation and abbreviations

Experimental design

No Comment

Validity of the findings

No Comment

Additional comments

The language in the manuscript still needs further improvement. The authors may check the manuscript for grammatical errors, typos, and sentence formation.
VFA production may also be taken into consideration during two-stage AD of MSW.

Reviewer 4 ·

Basic reporting

- Please emphasize the significant of the current study;
- The introduction section should refer to the recent reports (5 years);
- What is the significant of microbial analysis?
- Two-stage AD requires higher energy consumption comparing to single-stage AD, thus, It would be more convinced if the authors calculate the energy balance for both cases;
- It seems there was a positive correlation between organic loading and alkalinity at IFA 1%, but not with IFA 0.5%. Can you explain that?
- Please provide the high quality figures (≥300 dpi);
- Please provide the initial pH of the tests;
- Please provide the characteristics of inoculum;
- What are the components of the high moisture MSW in the current study?

Experimental design

- Experimental design should be complied in accordance with a standard test method such as ASTM D5511-12;

- High organic loading points may require higher ash addition to attain the suitable initial pH range, therefore, the ash addition at higher percentages should be investigated.

Validity of the findings

no comment

Additional comments

The present study falls within the scope of PeerJ and covers an interesting and important area of research. The manuscript has clear content and tight structure. However, there are some weak points that make the current manuscript unsuitable for publication.

---

## Round 0.3 · accepted · Accept

The manuscript is accepted if the authors agree with the removal of the sentence "Municipal solid waste (MSW) management using the incineration method generates ash and high moisture MSW as residue" from the background session in the Abstract, since it is unnecessary and confusing.